# MeCP2 regulates Tet1-catalyzed demethylation, CTCF binding, and learning-dependent alternative splicing of the *BDNF* gene in Turtle

Zhaoqing Zheng, Ganesh Ambigapathy, Joyce Keifer*

Neuroscience Group, Basic Biomedical Sciences, University of South Dakota, Sanford School of Medicine, Vermillion, United States

**Abstract** *MECP2* mutations underlying Rett syndrome cause widespread misregulation of gene expression. Functions for MeCP2 other than transcriptional are not well understood. In an ex vivo brain preparation from the pond turtle *Trachemys scripta elegans*, an intraexonic splicing event in the brain-derived neurotrophic factor (*BDNF*) gene generates a truncated mRNA transcript in naïve brain that is suppressed upon classical conditioning. MeCP2 and its partners, splicing factor Y-box binding protein 1 (YB-1) and methylcytosine dioxygenase 1 (Tet1), bind to *BDNF* chromatin in naïve but dissociate during conditioning; the dissociation correlating with decreased DNA methylation. Surprisingly, conditioning results in new occupancy of *BDNF* chromatin by DNA insulator protein CCCTC-binding factor (CTCF), which is associated with suppression of splicing in conditioning. Knockdown of MeCP2 shows it is instrumental for splicing and inhibits Tet1 and CTCF binding thereby negatively impacting DNA methylation and conditioning-dependent splicing regulation. Thus, mutations in *MECP2* can have secondary effects on DNA methylation and alternative splicing.

*For correspondence: jkeifer@usd.edu

**Competing interests:** The authors declare that no competing interests exist.

## Introduction

Rett syndrome (RTT) is an X-linked neurodevelopmental disorder caused by mutations in the methyl-CpG-binding protein 2 (*MECP2*) gene (*Amir et al., 1999*; *Bienvenu and Chelly, 2006*; *Lyst and Bird, 2015*). Loss of MeCP2 function in RTT leads to progressive neurological dysfunction including severe intellectual disabilities. MeCP2 binds to 5-methylcytosine (5mC) at CG and non-CG dinucleotides, particularly methylated CA, and 5-hydroxymethylcytosine (5hmC; *Mellén et al., 2012*; *Chen et al., 2015*; *Gabel et al., 2015*; *Kinde et al., 2015*) across the genome. Mutations that give rise to RTT result in widespread misregulation of genes consistent with a role in transcriptional regulation (*Lyst and Bird, 2015*). Additional mechanisms of action apart from transcription have also been proposed. Recent evidence indicates that MeCP2 also directly suppresses microRNA processing affecting dendritic growth and possibly neural development (*Cheng et al., 2014*). However, other functions are less well understood. One example is a role for MeCP2 in regulating alternative splicing.

Alternative splicing of pre-mRNAs of active genes is ubiquitous in the eukaryotic genome. It is estimated that as many as 90% of genes are affected by alternative splicing and its importance in brain has recently received a great deal of attention (*Scheckel et al., 2016*; *Zhang et al., 2016*). However, details of the molecular mechanisms that generate mRNA splice variants, under what conditions they are expressed in the nervous system, and the potential function of different protein isoforms are not well characterized. With respect to MeCP2 function, *Young et al. (2005)* found that MeCP2 partners with the Y-box binding protein 1 (YB-1) that binds DNA or RNA and is involved in

alternative splicing events (*Matsumoto and Wolffe, 1998*; *Gonzales et al., 2012*). More recently, MeCP2 was found to associate directly with other splice regulatory factors and, importantly, mutations in *MECP2* that occur in RTT disrupt these interactions (*Long et al., 2011*; *Li et al., 2016*). These studies suggest that aberrant alternative splicing resulting from a loss of MeCP2 may be an important contributor in RTT. MeCP2 is a highly phosphorylated protein and mutations at phosphorylated sites cause RTT-like behaviors in mice (*Ebert et al., 2013*; *Lyst et al., 2013*). However, whether functions of MeCP2, including splicing abnormalities, would be evident at steady state or mainly in response to neuronal activity is, surprisingly, an open question.

To approach the question of whether MeCP2 may regulate alternative splicing under physiological conditions in an intact neuronal circuit, we exploited a neural correlate of eyeblink classical conditioning using an isolated preparation from the turtle brain (*Keifer and Houk, 2011*; *Zheng et al., 2012*). The advantage of the preparation is that behaviorally relevant nerve-specific stimulation is used in a conditioning paradigm rather than non-specific stimuli such as glutamate application to induce a neural correlate of learning. Paired nerve stimulation evokes a behavioral, albeit 'fictive', physiological response that mimics features of conditioning in behaving animals. This model system allows the study of rapid learning-dependent epigenetic modifications in neurons that directly generate the learned behavior. We focused on the highly conserved growth factor brain-derived neurotrophic factor (*BDNF*) which is an activity regulated gene required for associative learning. The *BDNF* gene is an established target of MeCP2 in turtle as it is in humans and rodents (*Amir et al., 1999*; *Bambah-Mukku et al., 2014*; *Ambigapathy et al., 2015*) and *BDNF* transcripts are alternatively spliced (*Liu et al., 2005*; *Aid et al., 2007*; *Pruunsild et al., 2007*; *Ambigapathy et al., 2013*). We previously determined that splicing of the *BDNF* gene in turtle (*tBDNF*) results in a 40 base pair (bp) deletion within the protein coding sequence that generates a novel truncated *BDNF* transcript (*tBDNF2a*) and protein (*Ambigapathy et al., 2013*). Importantly, this intraexonic splicing event is regulated by conditioning such that splicing occurs in the naïve, untrained state and is rapidly and completely suppressed during training. Moreover, suppression is stimulus-specific. Splicing is inhibited by paired, but not unpaired, nerve stimulation and not by general increases in neuronal excitability evoked by application of glutamate or high potassium (*Ambigapathy et al., 2013*). The functional significance of this event in conditioning is, however, unknown.

The present study examines the role of MeCP2 in this *tBDNF* intraexonic splicing event and its rapid suppression during classical conditioning (*Ambigapathy et al., 2013*). In naïve preparations, MeCP2 binds to methylated sites in DNA upstream of the *tBDNF* splice site to yield truncated *tBDNF2a*. During conditioning, the site undergoes dynamic loss of 5mC and 5hmC, MeCP2 dissociates, and *tBDNF2a* splicing is suppressed. We identify the splicing factor, YB-1, and surprisingly, the methylcytosine dioxygenase 1 (Tet1) protein, in complexes with MeCP2 and show binding of all three to the same region of *tBDNF* in the naïve state. Furthermore, knockdown of MeCP2, YB-1 or Tet1 inhibits splicing of *tBDNF2a*. Demethylation and loss of MeCP2 during conditioning are correlated with binding of the DNA insulator protein CCCTC-binding factor (CTCF), leading to the idea that CTCF suppresses splicing. These findings suggest a new perspective for understanding mechanisms underlying RTT; that MeCP2 loss of function generates unexpected secondary deficits in patterns of gene methylation that results both in aberrant binding by DNA regulatory proteins and in alternative splicing. Further, and importantly, we show that the deficits are exposed in a physiological learning-dependent context, conditioning, and not under steady state conditions.

## Results

### Conditioning-dependent alternative splicing of *tBDNF2a* transcript is mediated by MeCP2

The vertebrate *BDNF* gene consists of a number of unique non-coding 5′ exons each having its own promoter that regulates a common 3′ protein coding exon with a variable 3′ untranslated region (UTR) that codes for expression of a preproBDNF precursor protein. Therefore, the *BDNF* gene encodes two-part transcripts having one non-coding exon spliced to the common protein coding sequence. After intracellular processing, proBDNF undergoes proteolytic conversion to form the mature BDNF protein involved in numerous cellular functions including cell growth, synapse formation and learning (*Lu et al., 2005*). We previously identified three non-coding 5′ exons (exons I-III)

and their promoters and one 3' protein coding exon (exon IV) for the *tBDNF* gene in turtle (*Ambigapathy et al., 2013*, *2014*). Exon II generates four mRNA transcripts designated *tBDNF2a-d* each having a short, intermediate, or long 3'UTR (*Figure 1A*).

The alternatively spliced *tBDNF2a* transcript encoding the truncated BDNF protein has an intermediate 3'UTR like *tBDNF2c* (*Figure 1A*) but is generated by an intraexonic splicing event in coding exon IV that removes 40 bp (13 amino acids) out of the distal sequence (*Figure 1B*) that we described previously in relation to conditioning (*Ambigapathy et al., 2013*). This deletion produces a frame shift resulting in an early stop codon and unique C-terminal end (*Figure 1B*). Further analysis here using primers that spanned the site of the deletion confirmed that this was an unconventional splicing event within the coding exon. The results from PCR of naïve samples showed a single band from genomic DNA at 225 bp (*Figure 1C*). By comparison, using cDNA as a template, two bands were obtained, one at 225 bp and a band that is 40 bp smaller at 185 bp. Sequencing verified that the larger band is from the *tBDNF2b-d* transcripts which are identical in this region of the coding sequence and the smaller band is *tBDNF2a* generated from the atypical 40 bp deletion (*Figure 1C*; *Supplementary file 1*). Throughout this report we will refer to this event for *tBDNF2a* as 'splicing' even though it is not typical of splicing generally.

All four *tBDNF* exon II transcripts are expressed in naïve, untrained preparations (*Figure 1D*). However, after 15 min of conditioning all of the transcripts are significantly downregulated (*Figure 1D,C15*) as was reported previously (*Ambigapathy et al., 2013*). The *tBDNF2a* transcript, specifically, is nearly completely suppressed to just 8% of naïve values after conditioning which is shown quantitatively in *Figure 1E* (n = 5/group, p=0.001, N vs. $C_{15\ min}$). Since MeCP2 has been strongly implicated in alternative splicing (*Young et al., 2005*; *Maunakea et al., 2013*), we examined whether an siRNA targeting *MECP2* affected splicing of the *tBDNF2a* transcript. We designed the siRNA based on conserved sequences from turtle DNA that correspond with human exon 4. The siRNA resulted in knockdown of total MeCP2 protein expression to an average of 59% of normal naïve levels (*Figure 1F*, n = 4/group, p<0.001). Significantly, bath application of the MeCP2 siRNA to naïve preparations resulted in substantial inhibition of the *tBDNF2a* transcript as shown in *Figure 1D* (arrow) that was on average about 40% of naïve values (*Figure 1E*, p=0.001, N vs. MeCP2 siRNA). A scrambled negative control siRNA (Silencer #1, Ambion) showed no reduction in expression of the spliced *tBDNF2a* transcript compared to normal naïve (*Figure 1D–E*, n = 2). Importantly, the other *tBDNF* transcripts, *2b-d*, were not inhibited by application of the MeCP2 siRNA (*Figure 1G*; n = 5/group, p<0.05, N vs. *tBDNF2c*; p=0.43, N vs. *2b* and *2d*) showing that it was selective in reducing *tBDNF2a*. These data support our interpretation that inhibition of the *tBDNF2a* transcript after MeCP2 siRNA is by suppression of a splicing mechanism and not by transcriptional suppression. Together, these data provide strong evidence for MeCP2-mediated intraexonic alternative splicing of the *tBDNF* coding sequence in naïve preparations.

## Suppression of *tBDNF* splicing during conditioning involves demethylation and release of MeCP2 binding

Since MeCP2 mediated splicing of the truncated *tBDNF2a* transcript, we used bisulfite sequencing PCR (BSP) to analyze the conditioning-dependent methylation status of the *tBDNF* protein coding sequence (*Table 1*). A schematic illustration of the protein coding sequence shows the relative position of CG and CA sites whose methylation status is altered by conditioning; the region deleted by splicing is indicated by the green box (*Figure 2A*, upper panel). In naïve preparations, nearly all of the CG sites are methylated (*Figure 2A*, middle panel). After only 15 min of conditioning, they undergo active demethylation ($F_{(1,82)}$ = 63.8, p<0.0001, N vs. $C_{15\ min}$) that does not occur following pseudoconditioning trials of randomly unpaired stimuli ($F_{(1,82)}$ = 0.15, p=0.70, N vs. Ps). We also observed that there are selected CA sites in the coding sequence that are either methylated in naïve and demethylated after conditioning (e.g., site 26, p=0.002) or methylated after conditioning (*Figure 2A*, lower panel, e.g., sites 27, 39, 40, 46, p<0.01). Therefore, in addition to CG dinucleotides, CA sites in the coding region also undergo active conditioning-dependent methylation/demethylation in only 15 min. To evaluate whether demethylation has a causative role in cessation of splicing of the *tBDNF2a* transcript, we chemically induced demethylation by incubating preparations in the DNMT inhibitors zebularine or RG108. This treatment results in nearly complete inhibition of *tBDNF* methylation levels to about 10% (*Ambigapathy et al., 2015*). Splicing of the *tBDNF2a* transcript was significantly inhibited after zebularine or RG108 application compared to naïve and

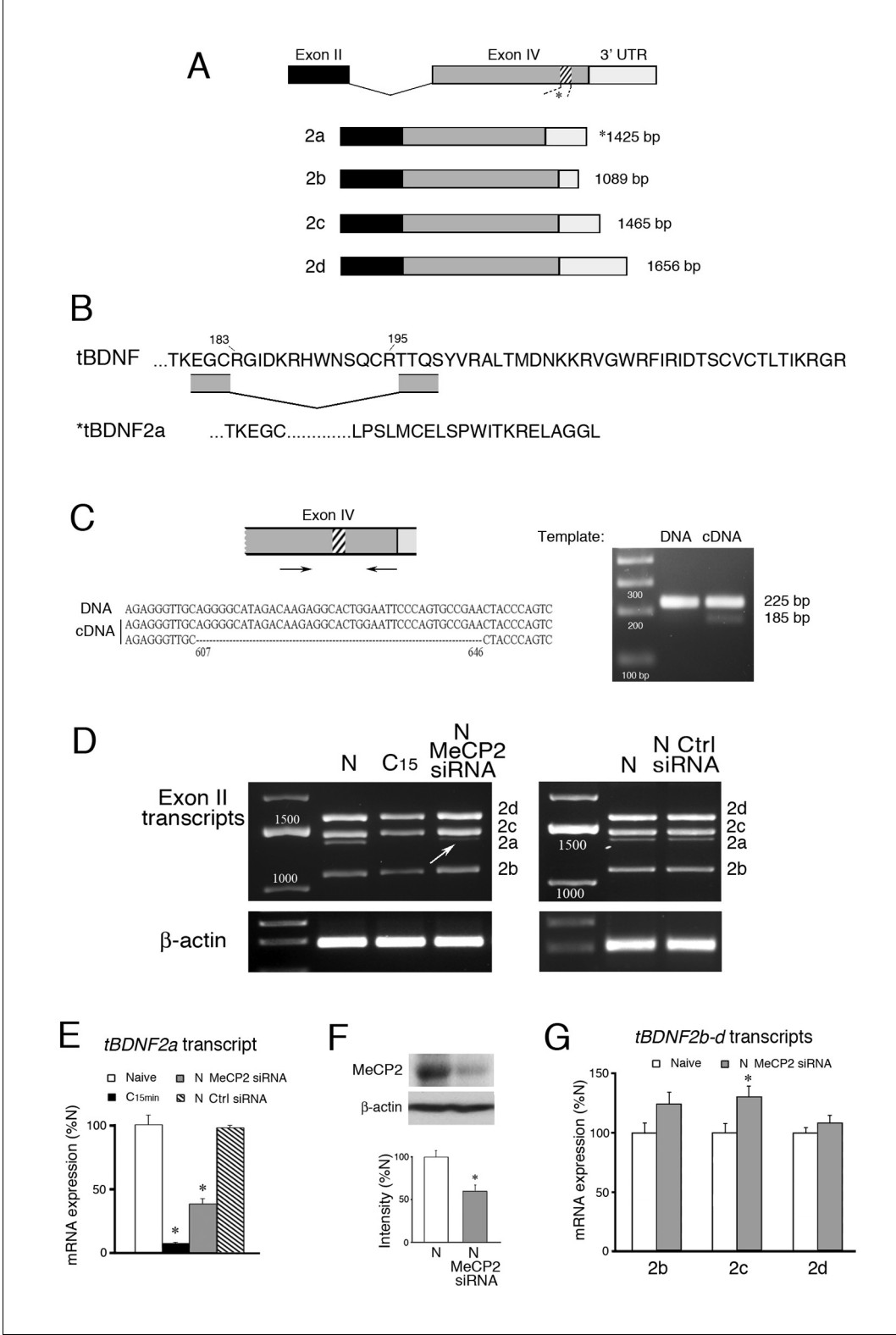

**Figure 1.** Conditioning-dependent alternative splicing of *tBDNF* is mediated by MeCP2. (**A**) Schematic diagram of *tBDNF* mRNA transcripts generated from non-coding exon II and the protein coding exon IV. Four transcripts designated *tBDNF2a-d* are produced in naïve preparations but only the *tBDNF2a* transcript undergoes an intraexonic splicing event in which 40 bp (13 amino acids) are removed from the distal region of the coding sequence (shown by the *hatching* at nt 607–646). (**B**) The amino acid sequence of the distal end of the protein

*Figure 1 continued on next page*

*Figure 1 continued*

coding sequence is shown for the full-length *tBDNF* transcripts. The deletion of the 13 amino acids in *tBDNF2a* (aa 183–195) is also shown which results in a frame shift and alternative C-terminal end with an early stop codon that generates the truncated tBDNF protein. Complete sequences of the tBDNF and tBDNF2a preproBDNF proteins are shown in *Ambigapathy et al. (2013)*. (C) The region of exon IV that undergoes the splicing event was further analyzed using primers flanking the splice site. The PCR products generated from genomic DNA produced a single band at 225 bp while cDNA produced two bands at 225 bp and 185 bp. Sequencing showed that the larger PCR band from cDNA was identical to *tBDNF2b-d*, while the smaller band was *tBDNF2a* (accession numbers: KC151267 – KC151270). (D) Four *tBDNF* exon II transcripts *2a-d* are expressed in naïve (N) preparations. After 15 min of conditioning (C₁₅), all transcripts are downregulated but only the *2a* transcript is nearly completely suppressed. Application of a MeCP2 siRNA (200 nM, 24 hr) to naïve preparations inhibits *tBDNF2a* expression (*arrow*) while a control siRNA (Ctrl siRNA; 200 nM, 24 hr) does not. (E) Semi-quantitative data of *tBDNF2a* mRNA expression in the different experimental conditions is shown relative to naive. The *tBDNF2a* transcript is significantly reduced during conditioning and after treatment with MeCP2 siRNA. (F) Western blots confirm that the MeCP2 siRNA significantly inhibits total MeCP2 protein compared to normal naïve. (G) Expression of the remaining exon II transcripts, *tBDNF2b-d*, is not inhibited by application of MeCP2 siRNA. For this and all figures, p and n values are given in the text.

pseudoconditioned preparations (Fig. 2B, $F_{(3,16)}$ = 80.0, n = 5/group, p<0.01). These data indicate a strong mechanistic role for demethylation in the cessation of splicing.

To establish a conditioning-dependent association between splicing and MeCP2 binding to *tBDNF*, we analyzed selected regions of the coding sequence with restricted primer sets using ChIP-qPCR assays. Three sets of primers were used to survey the region, P1-P3 (*Table 2*), whose relative coverage is indicated in the schematic diagram of the *tBDNF* coding sequence (*Figure 2A*). The CG and CA sites covered by each primer set are also indicated in *Figure 2A*. ChIP-qPCR showed strong binding of total MeCP2 protein to *tBDNF* in the naïve state corresponding with splicing of the *tBDNF2a* transcript (*Figure 2C*, 2a_ON). This was observed in the region surveyed by primer set P1 which is, interestingly, upstream of the actual splice site. With conditioning when the coding sequence undergoes rapid demethylation, ChIP indicates a significant reduction in MeCP2 binding to just 4% of naïve (*Figure 2C*, n = 5/group, p<0.0001, N vs. C₁₅ ₘᵢₙ) that corresponds with cessation of splicing and suppression of the *tBDNF2a* transcript (2a_OFF). During this time, MeCP2 was also observed to increase binding in the region surveyed by primer set P2 (*Figure 2C*, p=0.07) suggesting that it may be translocated to this downstream site rather than vacating genomic DNA entirely. These alterations in binding occurred even though levels of total MeCP2 protein were unchanged during conditioning (*Figure 2D*; n = 4/group, p=0.50). Finally, ChIP-qPCR verified that MeCP2 binding was significantly reduced in naïve preparations exposed to MeCP2 siRNA to 35% of normal naïve levels (*Figure 2C*, p<0.001, N vs. siRNA) corresponding well with the 40% reduction in the *tBDNF2a* transcript after MeCP2 siRNA (*Figure 1E*). These findings indicate that the *tBDNF* protein coding sequence is rapidly demethylated during conditioning resulting in reduced MeCP2 binding upstream of the splice site that corresponds with suppression of the alternatively spliced truncated *tBDNF2a* transcript.

**Table 1.** Primers used for BSP analysis of *tBDNF* coding sequence.

| Target | Orientation | Sequence (5'→ 3') |
|---|---|---|
| CG 1–4 | For | AAGTGTTAGAGGATTAGGTAGTTTGGTTTATTTAGGT |
| | Rev | AAAAACAATAAAAACTCCAAAAACAC |
| CG 2–11 | For | ATGTTATAGAGGAGTTTTTAGATGAGGA |
| | Rev | TACCTCTTATCTATACCCCTACAACCCTCTTT |
| CG 12–14 | For | AGAGGGTTGTAGGGGTATAGATAAGAGGTA |
| | Rev | CGATTCTTAACAACGACAACAAACCACAA |

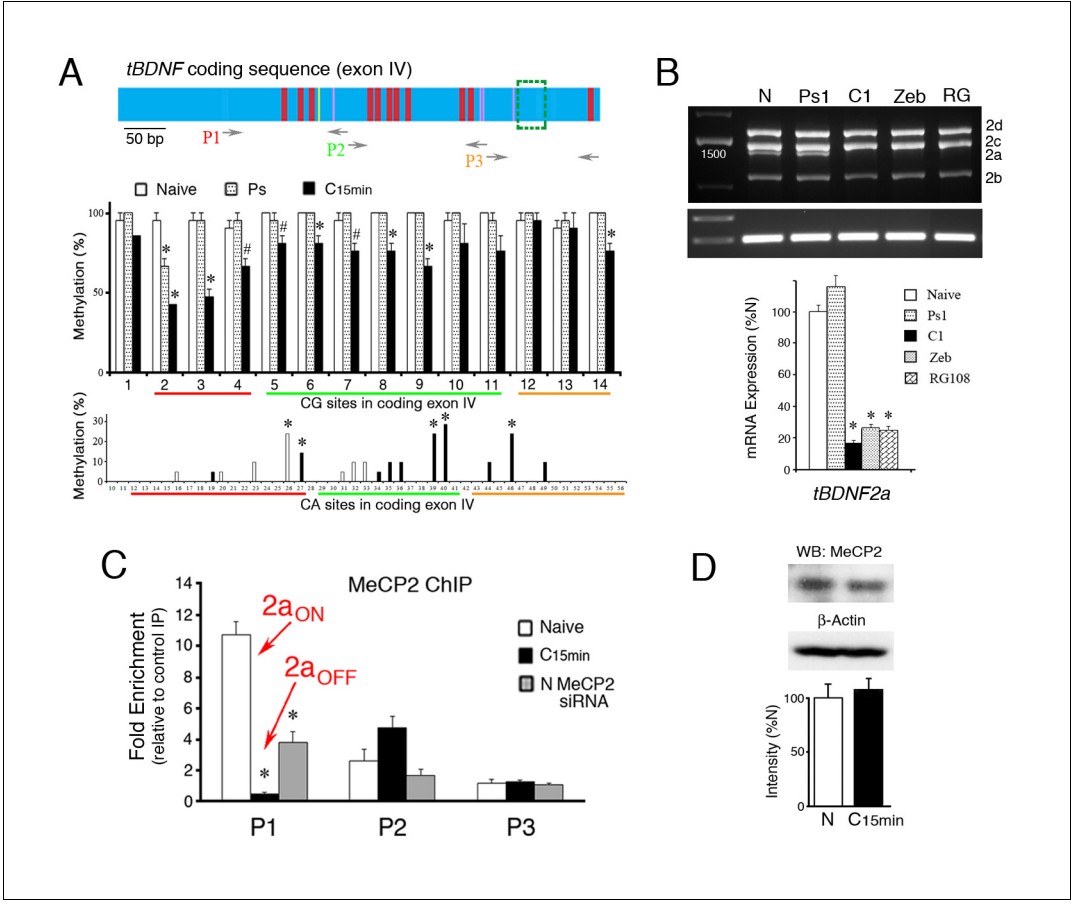

**Figure 2.** Suppression of *tBDNF* splicing during conditioning involves demethylation and release of MeCP2 binding. (A) Schematic illustration of the *tBDNF* protein coding sequence (upper panel) showing demethylated CG sites during conditioning in *red*, demethylated CA sites in *yellow*, and methylated CA sites in *purple*. The region deleted by splicing is indicted by the *green box*. The relative coverage of primer pairs P1-P3 used for ChIP-qPCR is indicated. The middle panel shows the methylation status of the *tBDNF* coding sequence determined by bisulfite sequencing PCR (BSP) in naïve (N), pseudoconditioned (Ps), and conditioned preparations (C15 min). Nearly all CG sites are significantly demethylated after conditioning for 15 min. 3 × 7 clones/group. As shown in the lower panel, specific CA sites also undergo conditioning-related methylation or demethylation. Bars in *white* are sites that are methylated in naïve and completely demethylated to 0% in conditioning; bars in *black* are unmethylated in naïve and methylated in conditioning. 3 × 10 clones/group. *Red*, *green*, and *yellow* lines indicate coverage by primer sets P1-P3, respectively. #p<0.05; *p<0.01. (B) Expression of the spliced *tBDNF2a* transcript is significantly inhibited relative to naïve by the DNMT inhibitors zebularine (Zeb; 100 µM) or RG108 (RG; 200 ng/µl) applied to the bath as shown by the semi-quantitative analysis. Levels of *tBDNF2a* transcripts from pseudoconditioned (Ps1) or conditioned (C1) preparations for one pairing session are also shown. (C) ChIP-qPCR assays for total MeCP2 protein show that primer set P1 detects MeCP2 tightly bound to *tBDNF* in naïve preparations during splicing of *tBDNF2a* (2aON) and is released after conditioning when splicing stops and *tBDNF2a* is suppressed (2aOFF). Application of the MeCP2 siRNA to naïve preparations results in significantly reduced ChIP signal in the P1 region compared to normal naïve. (D) Western blots confirm that levels of total MeCP2 are not altered during conditioning.

## YB-1 and Tet1 partner with MeCP2 and are required for *tBDNF* splicing

The DNA/RNA binding protein and splice factor YB-1 was previously reported to be a binding partner with MeCP2 (*Young et al., 2005*; *Gonzales et al., 2012*). To determine if this is also the case in our model system, co-immunoprecipitations (co-IPs) of MeCP2 and YB-1 were performed. The results show a strong interaction of total MeCP2 protein with YB-1 in both naïve and conditioned preparations that was not altered by conditioning (*Figure 3A*, n = 3/group, p=0.91). Additionally,

**Table 2.** Primers used for ChIP and MeDIP assays of *tBDNF* coding sequence.

| Target | Orientation | Sequence (5'→ 3') |
| --- | --- | --- |
| P1 | For | AGCCTAAGTGGGCCCAACAC |
| | Rev | TCCTCAAGCAGAAAGAGCAATG |
| P2 | For | GATGCTGCAAATATGTCCATGAG |
| | Rev | TCAGTTGGCCTTTGGGTACTG |
| P3 | For | CAAATGCAATCCCAAAGGTTACACAA |
| | Rev | TCTTATAAACCGCCAGCCAACT |

we confirmed that the MeCP2-YB-1 association required the presence of RNA. Treatment of samples with DNaseI did not affect the co-IPs whereas treatment with RNase A inhibited it (*Figure 3B*). Therefore, there is a strong interaction between MeCP2 and YB-1 that requires RNA. Evidence in favor of YB-1 involvement in splicing was obtained by using ChIP-qPCR. Since ChIP showed a significant relationship between MeCP2 binding to *tBDNF* and splicing in the coding sequence surveyed by primer set P1, we concentrated on examining this region for splice-related events. Like MeCP2, ChIP indicated that YB-1 was bound to this region in naïve preparations and was significantly reduced after conditioning (*Figure 3C*, n = 5/group, p=0.02). Importantly, western blot data showed that the overall level of YB-1 protein was not altered after 15 min of conditioning compared to naive (n = 3/group, p=0.82). These results indicate that YB-1 binds to the same region of the coding sequence as MeCP2 during splicing and is similarly released in conditioning when it is suppressed.

We previously showed that Tet1 controls DNA regulatory protein binding and histone modifications at *tBDNF* promoter regions (*Ambigapathy et al., 2015*). Moreover, Tet1 was shown to interact with MeCP2 in a conditioning-dependent manner. In this study, we observed significant demethylation of the coding sequence during conditioning. Since Tet1 is actively engaged in this process, we examined its role in the regulation of *tBDNF* methylation status during splicing. ChIP-qPCR showed that Tet1 is bound to the coding sequence in naïve and dissociates after conditioning (*Figure 3C*, p<0.001, N vs. C$_{15\ min}$). Taken together, the ChIP data show that MeCP2, YB-1 and Tet1 are bound to a region upstream of the splice site in naïve preparations during splicing of the truncated *tBDNF2a* transcript and all three dissociate during conditioning when splicing stops. Interestingly, release of binding during conditioning corresponds with changes in histone modifications in the same region of the coding sequence that consists of significantly increased levels of H3K4me3 (*Figure 3C*, p=0.01) and reduced H3K27me3 (p<0.01, N vs. C$_{15\ min}$), alterations that would lead to a more permissive, or open, chromatin structure.

Whether YB-1 has a direct role in splicing of the *tBDNF2a* transcript in naïve preparations was examined using a commercially prepared YB-1 siRNA. Bath application of the siRNA to naïve preparations resulted in a significant 53% knockdown of YB-1 protein expression compared to normal naïve (*Figure 3D*, n = 3/group, p<0.01). Significantly, analysis of mRNA transcripts in naïve preparations treated with YB-1 siRNA showed that expression of spliced *tBDNF2a* was reduced to 42% of its normal naïve values (*Figure 3E*, n = 5/group; p=0.001). These findings were specific to the *2a* transcript as expression of the other three exon II transcripts were not significantly affected by YB-1 siRNA treatment (*tBDNF2b-d*, $F_{(1,28)}$ = 0.10, p=0.75). Analysis of transcripts after application of a Tet1 siRNA to naïve preparations was previously reported in *Ambigapathy et al. (2015)*. These data are replotted here and show that Tet1 siRNA applied to naïve nearly completely suppressed the *tBDNF2a* transcript to only 4% of normal naïve values (*Figure 3E*, p=0.0003). Therefore, YB-1 and Tet1 are key partners with MeCP2 that are required for alternative splicing of the *tBDNF2a* transcript.

## YB-1, Tet1 and histone binding to *tBDNF* are suppressed by MeCP2 siRNA

Since MeCP2 may recruit YB-1, we examined whether MeCP2 siRNA inhibits YB-1 binding to the *tBDNF* coding region. Assessment of YB-1 ChIP revealed that it was significantly inhibited from binding in naïve preparations treated with MeCP2 siRNA (*Figure 3C*, p=0.01, N vs. siRNA) as would be

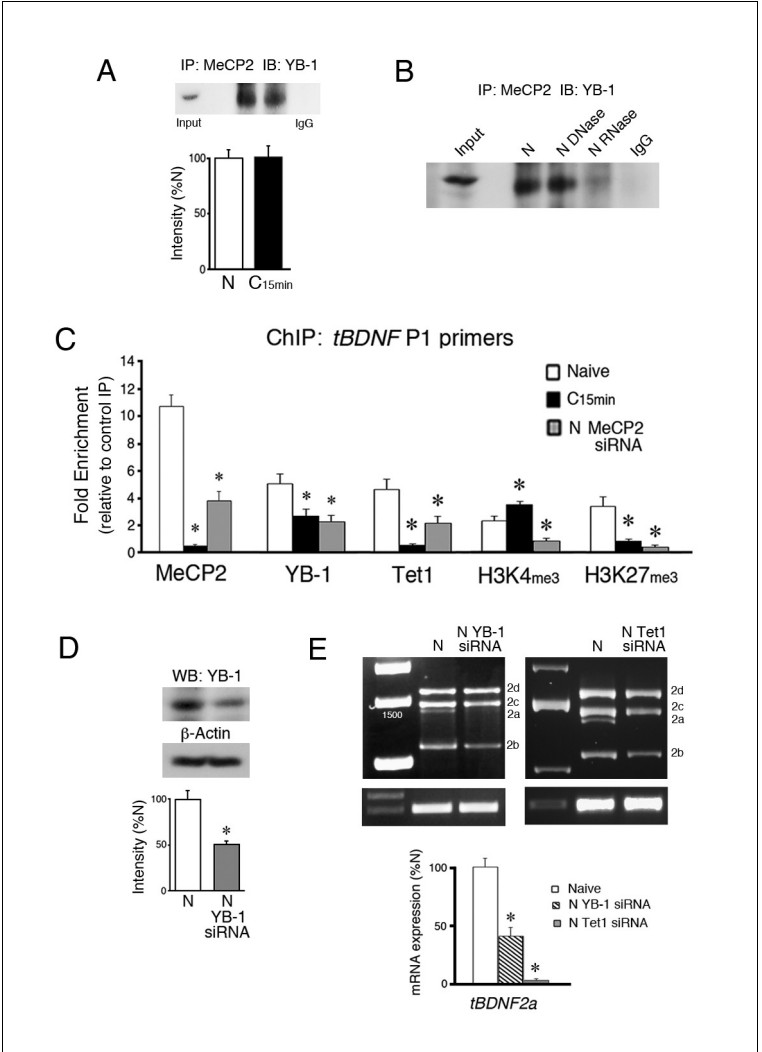

**Figure 3.** YB-1 and Tet1 partner with MeCP2 during splicing and are inhibited from *tBDNF* binding by MeCP2 siRNA. (**A**) Coimmunoprecipitation of MeCP2 with YB-1 shows a strong interaction in both naïve and conditioned preparations. A control co-IP is omitted because total MeCP2 protein immunoblots at a molecular weight similar to heavy chain IgG. IP, immunoprecipitation; IB, immunoblot. (**B**) Treatment of naïve brainstem samples with DNase had no effect on coimmunoprecipitation of MeCP2 with YB-1 while RNase treatment inhibited the interaction. (**C**) ChIP-qPCR assays of *tBDNF* binding in the region of the coding sequence surveyed by primer set P1. Data are from analysis of tissue samples from naïve preparations, those conditioned for 15 min ($C_{15\ min}$), and naïve treated with MeCP2 siRNA (N MeCP2 siRNA). *, Significant differences from naïve. (**D**) Western blots show that YB-1 siRNA (200 nM, 24 hr) significantly reduces expression of YB-1 protein to 53% of naïve. (**E**) The *tBDNF2a* transcript is significantly inhibited by siRNAs targeting either YB-1 or Tet1 applied to naïve preparations. Data for the Tet1 siRNA (150 nM, 24 hr) are replotted from *Ambigapathy et al. (2015)*.

expected if MeCP2 recruits YB-1. Because MeCP2 also has a transcriptional function, control experiments were performed showing that the MeCP2 siRNA did not alter the overall level of YB-1 protein compared to normal naïve (n = 3/group, p=0.66). Importantly, we also observed that Tet1 ChIP was significantly reduced by application of MeCP2 siRNA to naïve preparations (*Figure 3C*, p=0.02, N vs. siRNA). This occurred even though protein levels of Tet1 were also unaffected by the treatment (n = 3/group, p=0.57). Further follow-up of these results indicated that the histone modifications analyzed here, H3K4me3 associated with active genes and H3K27me3 associated with repressed genes, showed significantly reduced ChIP signals following MeCP2 siRNA treatment (*Figure 3C*,

H3K4me3, p=0.001; H3K27me3, p<0.01, N vs. siRNA) suggesting that MeCP2 also exerts control over chromatin structure.

To confirm that the inhibition of Tet1 binding to *tBDNF* by MeCP2 siRNA has functional consequences for regulation of methylation status, BSP was performed. During conditioning, the coding sequence undergoes rapid demethylation. Our results further show that this was significantly inhibited following conditioning in MeCP2 siRNA as shown for individual CG sites in the coding sequence (*Figure 4A*). Grouped data show there was an overall reduction in conditioning-induced demethylation across the coding sequence with MeCP2 siRNA treatment during conditioning compared to normal conditioning (*Figure 4B*, left panel, p<0.0001, $C_{15\ min}$ vs. $C_{15}$ MeCP2 siRNA). This was observed particularly for CG sites 2–4 surveyed within primer set P1 and the same region bound by MeCP2, YB-1 and Tet1 during splicing (*Figure 4B*, right panel, p<0.0001, $C_{15\ min}$ vs. $C_{15}$ MeCP2 siRNA). Therefore, inhibition of Tet1 *tBDNF* binding by MeCP2 siRNA treatment severely attenuates the demethylation process that normally takes place during conditioning.

Tet1 functions to convert 5mC to its oxidative derivatives including 5hmC (*Figure 4C*; *Guo et al., 2011*; *Wu and Zhang, 2011*) which is considered to be a stable epigenetic mark (*Hahn et al., 2013*; *Kinde et al., 2015*) bound by MeCP2 (*Mellén et al., 2012*; *Gabel et al., 2015*). Since BSP fails to distinguish 5mC from 5hmC, and to gain a better understanding of the methylation status of the *tBDNF* coding region during conditioning and after MeCP2 siRNA treatment, we performed methylated DNA immunoprecipitation (MeDIP) using antibodies selective for 5hmC. Our results showed that levels of 5hmC were high in the coding regions surveyed by primer sets P1 and P2 in naïve preparations (*Figure 4D*). After conditioning, values for 5hmC MeDIP were substantially reduced in all three primer set regions (*Figure 4D*, n = 5/group, $F_{(5,29)}$ = 58.2, p=0.001, N vs. $C_{15\ min}$), presumably reflecting conversion by Tet-mediated iterative oxidation to unmethylated C (*Figure 4C*). Moreover, treatment of naïve preparations with MeCP2 siRNA that inhibited Tet1 *tBDNF* binding also resulted in a significant reduction in 5hmC (Fig. 4D, $F_{(5,29)}$ = 33.4, p=0.001, N vs. N MeCP2 siRNA) consistent with Tet1's role in establishing this epigenetic mark. These findings suggest that in conditions in which Tet1 function is compromised by MeCP2 siRNA, 5mC predominates in the naïve state in place of 5hmC (*Figure 4C*). This interpretation is consistent with the BSP data from siRNA treated naïve preparations showing levels of methylation similar to normal naïve (*Figure 4B*).

## Inhibition of Tet1 blocks MeCP2-YB-1 *tBDNF* binding and splicing

In addition to MeCP2 siRNA, data showed that a Tet1 siRNA also blocks splicing in naïve preparations (*Figure 3E*) but the mechanism for this was unclear. Analysis of the methylation status of the coding sequence after Tet1 siRNA application by BSP showed that it remained highly methylated similar to untreated naïve preparations (*Figure 5A*, p=0.17). Further analysis by MeDIP demonstrated that levels of 5hmC were nearly doubled after Tet1 siRNA treatment compared to normal naïve (*Figure 5B*, p=0.001). Surprisingly, binding of MeCP2 and YB-1 to the *tBDNF* coding sequence was significantly inhibited (*Figure 5C*, n = 5/group, MeCP2, p<0.01, YB-1, p<0.01), which accounts for the suppression of splicing in Tet1 siRNA treated naïve preparations. Because the loss of Tet1 might result in compensatory actions of the other Tet proteins, we examined Tet3 by ChIP. These data showed minimal binding in normal naïve preparations that was greatly increased after Tet1 siRNA treatment (*Figure 5C*, n = 3/group, p=0.001). The enhanced binding of Tet3 was not accompanied by any changes in protein expression (*Ambigapathy et al., 2015*). These ChIP data indicate that the MeCP2-YB-1 complex was not recruited to methylated DNA under conditions in which there is loss of Tet1, which is replaced by Tet3.

## CTCF protects *tBDNF* DNA from splicing to suppress *tBDNF2a* transcripts during conditioning

The DNA-binding protein CTCF has been shown to have an insulator and chromatin organizing function that prohibits opposing regulatory interactions from spreading into neighboring domains (*Ali et al., 2016*; *Hnisz et al., 2016*). CTCF binding is also modified by DNA methylation such that it is evicted by 5mC to control exon splicing in the lymphocyte *CD45* model gene (*Shukla et al., 2011*; *Marina et al., 2016*). To assess the possibility that CTCF may be involved in splicing mechanisms in the *tBDNF* coding sequence, a search of CTCF binding site databases was conducted (www.insulatordb.uthsc.edu/). Predicted matches of *tBDNF* sequences with known CTCF binding motifs were

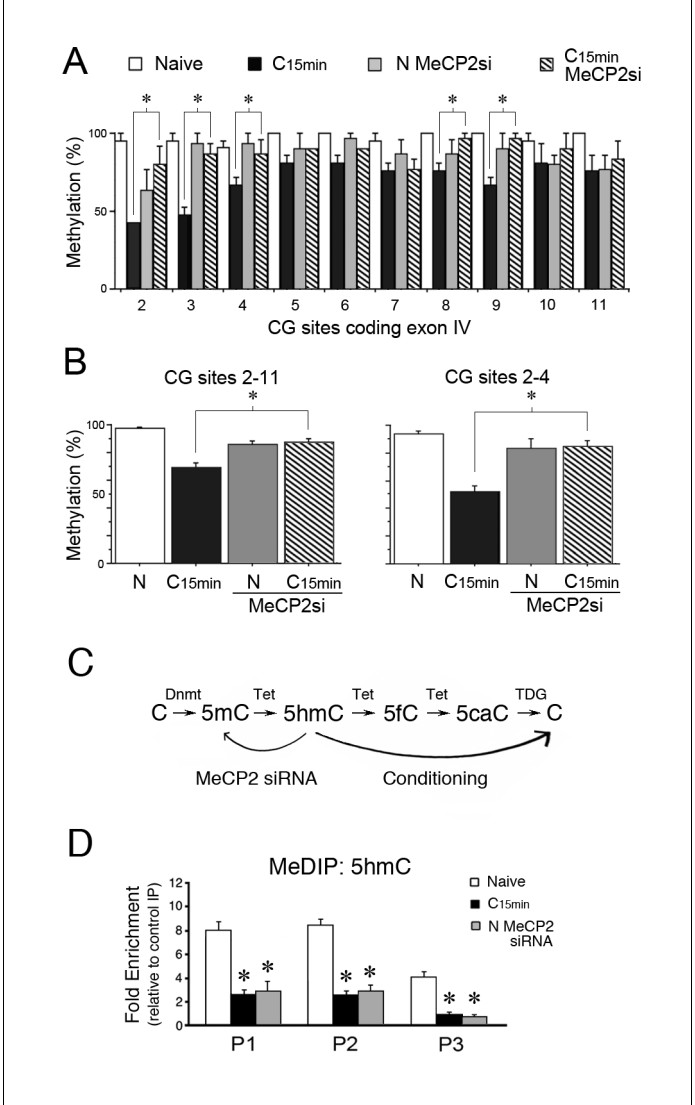

**Figure 4.** MeCP2 siRNA inhibits demethylation of the *tBDNF* coding sequence during conditioning. (**A**) BSP analysis of the coding sequence in naïve, conditioned preparations, and those treated with MeCP2 siRNA. Specific CG sites demethylated in normal conditioning show significantly greater levels of methylation after conditioning in MeCP2 siRNA. Data from normal naïve and conditioned are from *Figure 2* and are shown here for comparison. MeCP2 siRNA data are from $3 \times 10$ clones/group. *Significant differences (p=0.01) between the $C_{15\,min}$ MeCP2 siRNA treated group compared to normal $C_{15\,min}$. (**B**) Grouped data of CG sites 2–11 (left panel) and sites 2–4 (right panel) show significant attenuation of demethylation in conditioned preparations treated with MeCP2 siRNA compared to normal conditioning. (**C**) Schematic illustration of the putative stepwise oxidative demethylation pathway performed by Tet proteins and the thymine DNA glycosylase (TDG) and base excision repair system. Normal conditioning drives the process to the right toward unmethylated C, whereas treatment of naïve or conditioned preparations with MeCP2 siRNA that results in a reduction of Tet1 binding to *tBDNF* shifts the pathway to the left toward 5mC. 5fC, 5-formylcytosine; 5caC, 5-carboxylcytosine. (**D**) MeDIP assays show high levels of 5hmC content in the *tBDNF* coding sequence of naïve preparations. These levels are significantly reduced after normal conditioning and in naïve preparations treated with MeCP2 siRNA.

found for three regions. The first was located within primer set P1 immediately adjacent to CG site 2. The second match was placed within the region surveyed by primer set P2, and the third was centrally located within the splice site itself in the region covered by primer set P3. Since suggestive positive matches were found, we first evaluated whether CTCF protein expression was modulated

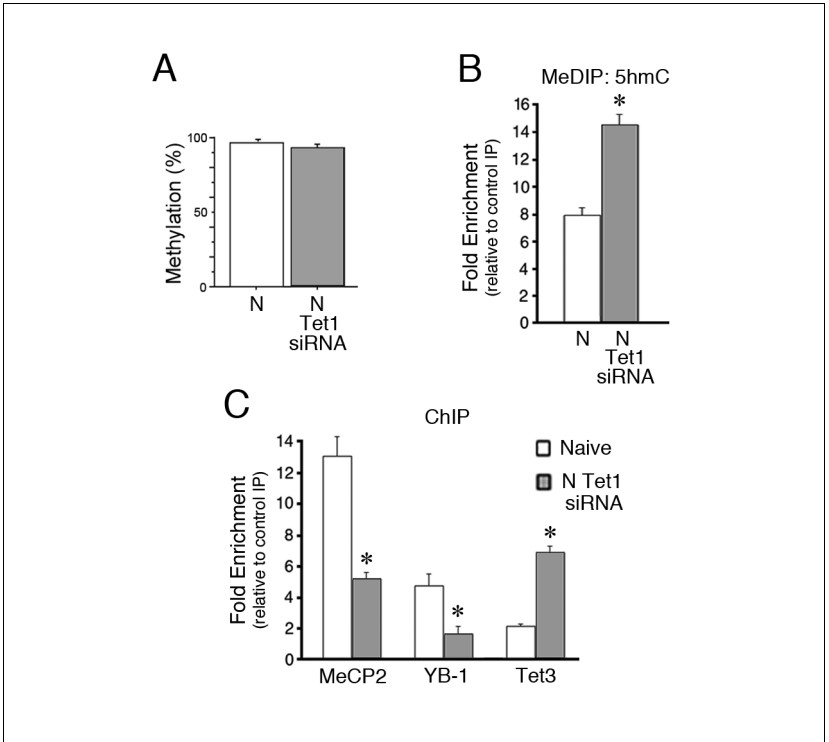

**Figure 5.** Inhibition of Tet1 by siRNA alters methylation and blocks MeCP2 and YB-1 binding to the coding sequence. (**A**) Methylation status of the *tBDNF* coding sequence (CpGs 2–11) in normal and Tet1 siRNA-treated naïve preparations analyzed by BSP. High levels of methylation were detected for both conditions. Normal naïve data are reshown here from earlier figures; Tet1 siRNA data are from 3 × 10 clones/group. (**B**) MeDIP assay showing the substantial increase in 5hmC in the P1 primer set region of the coding sequence after treatment of naïve preparations with Tet1 siRNA. Data from normal naïve are reshown from **Figure 4D**; n = 3 for Tet1 siRNA group. (**C**) ChIP assays reveal a significant reduction in binding of MeCP2 and YB-1 to the coding sequence at P1 after Tet1 siRNA treatment. In contrast, binding for Tet3 in conditions of Tet1 siRNA was dramatically increased. Data for YB-1 normal naïve are reshown from **Figure 3C** for comparison.

by conditioning. CTCF is highly conserved across species and migrates as a larger isoform at ~140 kDa in rat while the turtle protein is slightly smaller (**Figure 6A**). Expression of the CTCF protein was observed to be regulated during conditioning as it was significantly increased to 168% over naïve values (**Figure 6B**, n = 5/group, p=0.001). To assess a direct role for CTCF in *tBDNF2a* splicing, expression of mRNA transcripts in naive preparations and those treated with a CTCF siRNA was examined. The siRNA was designed against conserved CTCF sequences in the turtle (nt 1187–1212) that resulted in a reduction in protein expression to 27% of normal naïve preparations (**Figure 6C**, lower left panel, n = 3/group, p=0.001). Application of the siRNA to naïve preparations failed to have any effect on *tBDNF2a* splicing, as shown by the semi-quantitative data (**Figure 6C**, lower right panel, n = 5/group, p=0.91, N vs. N CTCF siRNA). Interestingly, however, when conditioned preparations that normally suppress splicing were treated with CTCF siRNA, the *tBDNF2a* transcript was unmasked and splicing resumed at levels similar to normal naïve preparations (**Figure 6C**, p<0.0001, $C_{15}$ vs. $C_{15}$ CTCF siRNA). These data provide compelling evidence that CTCF insulates DNA sequences from the splicing apparatus during normal conditioning thereby suppressing the truncated *tBDNF2a* transcript. This function was blocked by the CTCF siRNA during conditioning allowing splicing to take place. To provide additional support for our interpretation of splicing insulation, ChIP assays of CTCF binding to the coding region showed significantly increased signals within primer sets P1 and P3 during conditioning (**Figure 6D**, n = 5/group; P1: p=0.002; P3, p<0.05, N vs. $C_{15 \ min}$). While there was strong binding to region P2 this was not conditioning-dependent (**Figure 6D**, p=0.12). CTCF ChIP signals in the regions surveyed by P1-P3 therefore confirmed the

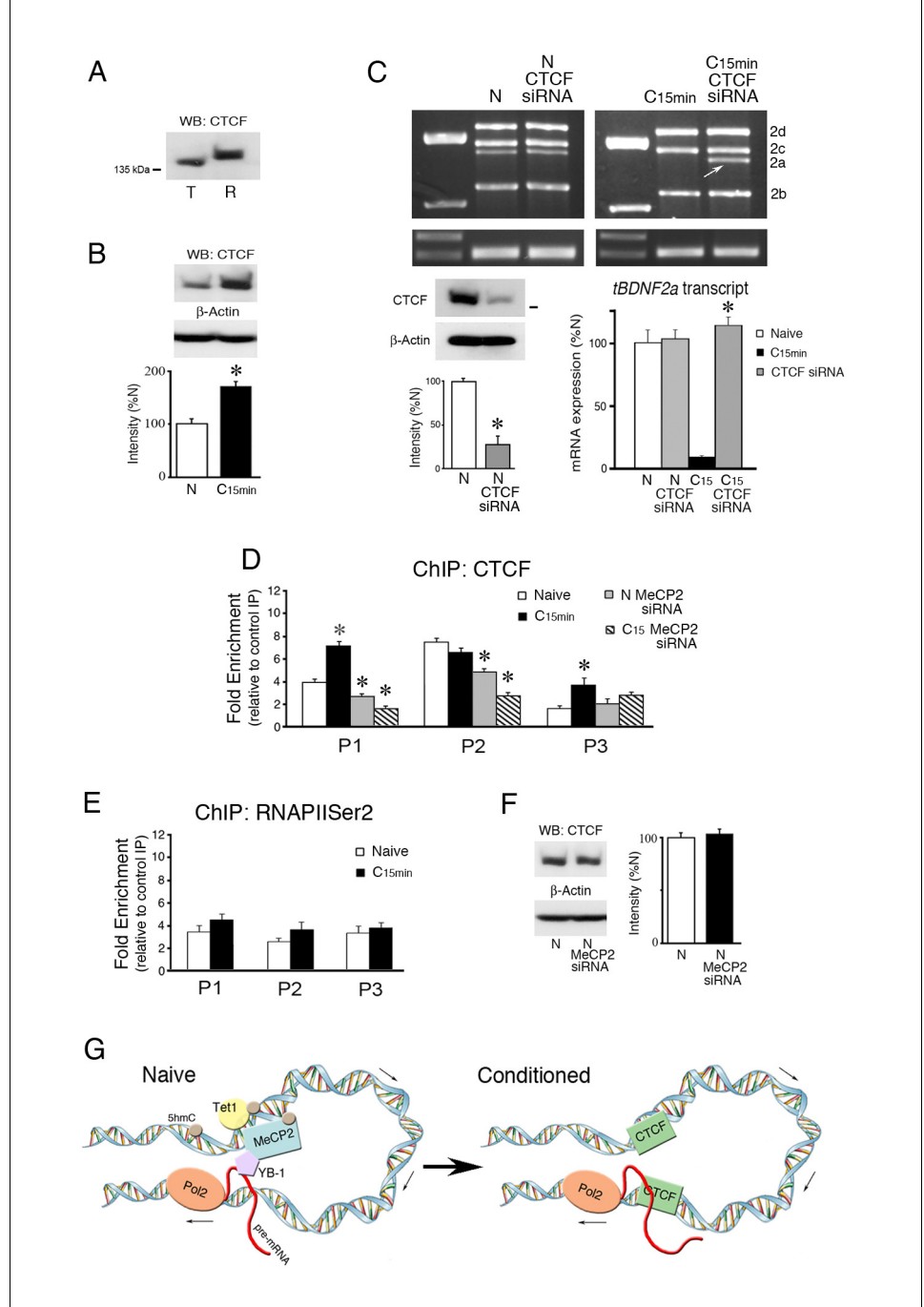

**Figure 6.** The insulator protein CTCF suppresses splicing of the *tBDNF2a* transcript during conditioning. (**A**) The CTCF antibody recognizes an ~140 kDa protein band in both turtle (T) and rat (R) brain tissue. (**B**) CTCF protein expression in turtle is significantly upregulated during conditioning. (**C**) Western blots show that the CTCF siRNA significantly inhibits CTCF protein expression compared to naïve (lower left panel). Expression of exon II transcripts after treatment with CTCF siRNA results in no change in the level of spliced *tBDNF2a* in treated naïve preparations, as shown by the PCR gels (upper panels) and semi-quantitative data (lower right panel). In conditioned preparations ($C_{15 \, min}$) in which *tBDNF2a* is normally suppressed, CTCF siRNA application results in strong expression of *tBDNF2a* (arrow) to near normal naïve levels. *p<0.0001, $C_{15}$ CTCF siRNA vs. $C_{15}$. (**D**) ChIP assays of CTCF show that binding to *tBDNF* is significantly increased after conditioning in regions surveyed by primers P1 and P3. Application of MeCP2 siRNA significantly reduces binding in both naïve (N) and conditioned ($C_{15}$) preparations in regions P1 and P2 compared to normal naïve. (**E**) ChIP of RNAPIISer2 binding to the coding exon shows no significant changes during conditioning compared to naïve. (**F**) Expression of CTCF protein is not

*Figure 6 continued on next page*

*Figure 6 continued*

affected by application of the MeCP2 siRNA. (**G**) Model summarizing the molecular events during splicing in naïve and conditioned preparations. The MeCP2 DNA binding region (surveyed by primer set P1) is upstream of the splice site (surveyed by primers P3). We hypothesize they are drawn into close proximity of one another by chromatin looping. In naïve preparations, DNA in the MeCP2 binding region is methylated with high 5hmC content maintained by Tet1 that colocalizes with MeCP2. MeCP2 recruits YB-1 that performs the splicing as the nascent transcript is elongated by RNAPII (Pol2) at the downstream leg of the loop to generate *tBDNF2a*. Upon conditioning, 5hmC is demethylated to unmodified C and Tet1 dissociates along with MeCP2 and YB-1. This is followed by new occupancy by CTCF that insulates DNA from further splicing thereby suppressing *tBDNF2a* during conditioning.

presence of the predicted CTCF binding sites. We next examined whether CTCF binding in conditioning interferes with RNA polymerase II (RNAPII) elongation as this has been previously implicated in splicing mechanisms (*Shukla et al., 2011*). Analysis with ChIP for RNAPIISer2 showed no significant conditioning-related changes in binding to the coding exon (*Figure 6E*, n = 5/group; $F_{(5,24)}$ = 1.40, p=0.26) suggesting that CTCF does not affect binding of the elongating form of RNAPII.

Finally, we tested whether MeCP2 siRNA has any effect on CTCF *tBDNF* binding as it did for Tet1 and YB-1. The results showed a significant reduction in CTCF ChIP of both naïve and conditioned preparations treated by MeCP2 siRNA compared to untreated preparations (*Figure 6D*, N siRNA P1: p=0.01, $C_{15}$ siRNA P1: p<0.0001; N siRNA P2: p<0.001, $C_{15}$ siRNA P2: p<0.001) even though overall protein levels were unaffected by the treatment (*Figure 6F*, n = 3/group, p=0.44). These data correspond well with our conclusion that MeCP2 siRNA inhibits Tet1-catalyzed demethylation during conditioning resulting in enhanced occupation of the coding region by 5mC and eviction of CTCF.

## Discussion

### *BDNF* gene structure and splicing across species

The *BDNF* gene is characterized by its complex structure and high level of diversity among species, having multiple promoters and non-coding exons to regulate the developmental timing, tissue specificity, and cellular localization of expression of a common protein coding sequence. Splicing donor and acceptor sites are used within the non-coding exons to generate unique *BDNF* transcripts. The functional significance of all of these diverse transcripts is not known. Notably, the human *BDNF* gene also codes for antisense *BDNF* transcripts that may additionally regulate transcript expression (*Liu et al., 2005*; *Pruunsild et al., 2007*) that may not be present in rat or mouse (*Aid et al., 2007*), although this is debatable (*Modarresi et al., 2012*). Therefore, the mammalian *BDNF* gene may be regulated in numerous ways to generate multiple transcripts, not all of which are characterized functionally. Although the intraexonic splicing event described here in turtle is an unconventional one, internal splicing, although rare, has been documented in other genes from normal human and patient populations (*Cox et al., 2001*; *Tran et al., 2006*; *Madrigal et al., 2016*) or rodents (*Ye et al., 2009*). While the functional significance of the spliced transcript in turtle *BDNF* is unknown, it generates a truncated protein that has a pattern of expression opposite to mature BDNF protein during conditioning (*Ambigapathy et al., 2013*), suggesting it may have a dominant negative function. In the present study, we use the conditioning-related intragenic splicing of the *BDNF* coding exon as a model event to characterize activity-dependent splicing related to MeCP2 function. Our evidence indicates that mechanisms regulating splicing, such as Tet-mediated DNA methylation and control of MeCP2-YB-1 and CTCF binding, are highly conserved and consistent with previous studies in mammals (*Young et al., 2005*; *Shukla et al., 2011*; *Maunakea et al., 2013*; *Marina et al., 2016*).

### MeCP2 regulates splicing and DNA methylation status indirectly through association with Tet1

Recent studies have linked DNA methylation to mechanisms of alternative splicing (*Shukla et al., 2011*; *Maunakea et al., 2013*; *Yearim et al., 2015*; *Marina et al., 2016*) and recruitment of MeCP2

binding (*Maunakea et al., 2013*) in non-neuronal cells. A genome-wide analysis reported that gene body methylation and MeCP2 binding are enriched in alternatively spliced exons and that inhibition of methylation disrupts selective targeting of MeCP2 resulting in aberrantly spliced transcripts (*Maunakea et al., 2013*). Our results are consistent with this observation. One proposed mechanism for MeCP2-mediated splicing is that MeCP2 recruits partner proteins that slow transcriptional elongation permitting more time for recognition of splice sites by the splicing machinery (*Shukla et al., 2011*; *Luco et al., 2011*; *Maunakea et al., 2013*). MeCP2 partners directly with splice factors including YB-1 whose actions directly result in pre-mRNA alternative splicing (*Young et al., 2005*; *Long et al., 2011*; *Gonzales et al., 2012*; *Li et al., 2016*). Therefore, accumulating evidence indicates that DNA binding by MeCP2 has a central role in splicing and is regulated by DNA methylation status. However, details of the underlying mechanisms are somewhat obscure and the activity-dependent interactions of MeCP2 with other DNA regulatory processes are scarcely characterized. During the *tBDNF* splicing event in naïve preparations, the MeCP2-YB-1 complex is bound to methylated DNA as is Tet1, as shown in our model (*Figure 6G*). Since the MeCP2 binding site (P1 region) is upstream from the splice site (P3 region), we propose that chromatin looping brings the two regions in close enough proximity to functionally interact. In this configuration, as the nascent mRNA transcript is elongated by RNAPII it is spliced by YB-1 to generate *tBDNF2a*. The mechanisms underlying YB-1 recognition of the splice site and how splicing is coordinated along with production of the other full-length exon II transcripts are unknown. Upon conditioning (*Figure 6G*), the *tBDNF* coding region undergoes demethylation, MeCP2 and partners YB-1 and Tet1 dissociate, and CTCF binds to insulate the splice site from further splicing. The interpretation that suppression of *tBDNF2a* during conditioning is by a splicing mechanism and not a transcriptional one is supported by data showing that the other three full-length transcripts (*2b-d*) are not inhibited by MeCP2 siRNA. Instead, their conditioning-induced downregulation is mediated by enhanced methylation of the exon II promoter and binding by the transcriptional repressor BHLHB2 (*Ambigapathy et al., 2015*).

Our findings further reveal that the interaction between MeCP2 and Tet1 is more complex than expected, and in this regard knockdown studies of each by siRNA are instructive. In the *tBDNF* coding sequence, both MeCP2 and Tet1 are bound at the same time in normal naïve preparations and both dissociate upon stimulation (*Figure 6G*). The presence of high levels of 5hmC in naïve could explain why Tet1 ChIP shows strong binding even though DNA is methylated suggesting it may function to maintain the 5hmC mark (*Hahn et al., 2013*; *Kinde et al., 2015*). Loss of MeCP2 in naïve preparations results in a reduction in Tet1 binding and levels of 5hmC in the coding sequence. A reduction in 5hmC with MeCP2 knockdown is corroborated by *Szulwach et al. (2011)*, but only for genomic loci in which 5hmC is dynamically regulated during development and not for those stably modified. They argued that stable loci might have redundant mechanisms using multiple methyl-CpG-binding domain (MBD) proteins that negatively regulate genomic 5hmC content (and presumably Tet binding) while dynamically regulated sites rely more heavily on MeCP2. Application of Tet1 siRNA, on the other hand, results in substantially increased levels of 5hmC, which has been observed by others using Tet1 knockdown in mammals (*Feng et al., 2015*). Moreover, and importantly, in conditions of Tet1 loss of function, DNA binding by MeCP2 was significantly reduced even though levels of methylation were high. Therefore, inhibition of either MeCP2 or Tet1 negatively impacts the function of the other leading to the conclusion that, at least for splicing regulation of *tBDNF*, MeCP2 and Tet1 are critical functional partners. Additionally, our data show that Tet3, which is expressed in substantial levels in turtle brain (*Ambigapathy et al., 2015*), binds to *tBDNF* with loss of Tet1 and likely enhances the deposition of 5hmC that was observed. While we confirmed our previous result that MeCP2-Tet1 co-IP in naïve preparations (*Ambigapathy et al., 2015*), preliminary findings show no MeCP2-Tet3 interaction (data not shown). Since MeCP2 *tBDNF* binding is reduced when Tet1 is replaced by Tet3, it is tempting to speculate that Tet3 does not recruit MeCP2 or may even repel it. The details of the MeCP2-Tet1 interaction remain to be fully characterized and may be different depending on their presence at gene promoters or coding regions, or stable and dynamically regulated genomic loci.

## CTCF protects DNA from splicing during conditioning

Our evidence indicates that a primary mechanism for splicing suppression is through conditioning-induced binding of the insulator protein CTCF. DNA binding by CTCF is repelled by the presence of 5mC while, in contrast, it preferentially binds to unmethylated C or possibly oxidized derivatives of

5mC (*Shukla et al., 2011*; *Spruijt et al., 2013*; *Marina et al., 2016*). Additionally, CTCF colocalizes to DNA domains enriched in active histone marks such as H3K4me3 or H3K36me3 and is generally excluded from regions with elevated repressive marks including H3K27me3 or H3K9me3 (*Saldaña-Meyer and Recillas-Targa, 2011*; *Ali et al., 2016*). We observed significantly elevated ChIP for CTCF in two regions of the *tBDNF* coding sequence after classical conditioning surveyed by primer sets P1 proximally and P3 located more distally (and within the splice site). This enhanced binding was associated with DNA demethylation and a reduction in 5hmC. Significantly, suppression of the spliced *tBDNF2a* transcript that normally occurs during conditioning was blocked by incubation with a CTCF siRNA, returning it to splicing levels similar to naïve preparations. These data provide compelling evidence that CTCF insulates *tBDNF* from the splicing machinery during conditioning thereby suppressing the spliced transcript.

Although the exact mechanism underlying conditioning-dependent CTCF protection of the *tBDNF* splice site has yet to be characterized, several possibilities can be offered based on previous work of others. *Shukla et al. (2011)* provided evidence that *CD45* exon inclusion was related to CTCF binding and RNAPII pausing. They showed that decreased ChIP for RNAPIISer2 and slower rates of transcription corresponded with increased presence of CTCF which was thought to act as a physical barrier to RNAPII and allow more time for splice site recognition (*Luco et al., 2011*). Our data showed no changes in *tBDNF* ChIP for RNAPIISer2 associated with CTCF suppression of splicing during conditioning. It is possible that CTCF has dual functions in regulating alternative splicing, through RNAPII pausing to promote it or by its insulator function to inhibit it. CTCF along with binding partner cohesin are also involved in chromatin looping that bends DNA located between paired binding sites of CTCF/cohesin (*Ali et al., 2016*). In consideration of the chromatin looping model, it is intriguing that two regions in the *tBDNF* coding sequence were observed to show significantly elevated CTCF binding after conditioning, a proximal region (P1) in which MeCP2-YB-1 binding/dissociation is correlated with splicing and a distal region (P3) located within the splice site itself. These may reflect paired CTCF/cohesin interactions required for loop formation that protects the *tBDNF* splice site from splicing. However, there are only 374 bp separating the P1 and P3 CTCF binding sites in the coding sequence and it is uncertain how short DNA loops occurring in nature can be or if short-range loops might be mediated by CTCF/cohesin. While loop or kink formation may be possible (*Pasi and Lavery, 2016*), whether these functionally insulate DNA is unknown.

## Impact of MeCP2 loss of function on DNA methylation and chromatin

MeCP2 protein is proposed to have a widespread impact on gene expression, both as an activator or repressor, chromatin structure, and alternative splicing (*Linhoff et al., 2015*; *Lyst and Bird, 2015*). Aberrant pre-mRNA alternative splicing is thought to be an important factor in generating the suite of abnormalities observed in patients with the neurodevelopmental disorder RTT and implicates MeCP2 in splicing mechanisms (*Young et al., 2005*; *Bienvenu and Chelly, 2006*; *Maunakea et al., 2013*; *Li et al., 2016*; *Cheng et al., 2017*). MeCP2 was reported to interact with YB-1 and Prpf3 (*Young et al., 2005*; *Long et al., 2011*; *Gonzales et al., 2012*), and other splicing factors (*Li et al., 2016*), but its exact role and the ubiquity of its actions in splicing are not well characterized. The data here confirm that YB-1 is a partner of MeCP2 and shows that YB-1 participates in the splicing of a truncated *tBDNF* transcript that is regulated during conditioning by modifications in DNA methylation. We also confirmed that the interaction of YB-1 with MeCP2 requires RNA further supporting its role in splicing. Moreover, we observed in ChIP assays that depletion of MeCP2 by siRNA results in deficits in *tBDNF* binding by Tet1 and other proteins involved in splicing including YB-1 and CTCF. This reduction in binding occurred without any changes in levels of protein expression. The loss of Tet1 binding in MeCP2-treated preparations resulted in a functionally significant elevation in 5mC in the *tBDNF* coding sequence that would normally undergo conversion to unmethylated C during conditioning. This change in methylation status was accompanied by inhibition of CTCF binding during conditioning that can be attributed, based on the work of others, to its eviction by abnormally high levels of 5mC. This outcome would be expected to result in reversal of splicing suppression during conditioning similar to the naïve state. Interestingly, in MeCP2 null male mice, dynamically modified, but not stable, 5hmC loci were found to be significantly reduced compared to controls (*Szulwach et al., 2011*), consistent with our findings. Taken together, available evidence suggests that depletion of MeCP2 protein may alter DNA methylation status indirectly by preventing binding of other DNA modifying proteins, including Tet.

How loss of MeCP2 binding results in a reduction in Tet1 binding to the *tBDNF* gene is unclear. Evidence indicates that MeCP2 and Tet1 are binding partners (*Cartron et al., 2013*; *Ambigapathy et al., 2015*) but the functional impact of this relationship has been largely unstudied. It is possible that MeCP2 and Tet1 together regulate the overall content of 5mC and 5hmC for dynamically modulated genes. Loss of MeCP2 destabilizes this regulatory process for control of DNA methylation across the genome. An alternative but not mutually exclusive mechanism relates to the view that MeCP2 is a global regulator of chromatin structure and may serve as an integral component of the chromatin architecture itself (*Skene et al., 2010*; *Cohen et al., 2011*). In our study, we observed that inhibition of MeCP2 severely depleted the *tBDNF* coding sequence of the histone modifications H3K4me3 and H3K27me3 that have a significant role in regulating gene expression. Therefore, loss of MeCP2 may affect learning-related *tBDNF* binding by regulatory proteins, including Tet1, through its action on controlling the accessibility of the surrounding chromatin structure. Along these lines, increased chromatin compaction observed in MeCP2-null neurons in vivo might restrict activity-dependent DNA regulatory mechanisms leading to a disruption in gene methylation and expression (*Linhoff et al., 2015*). Further work on the activity-dependent regulation of MeCP2, Tet1 and DNA methylation will be required to resolve their interactions and functional consequences on gene expression.

## Materials and methods

### Conditioning procedures

Freshwater pond turtles, *Trachemys scripta elegans*, of either sex were purchased from commercial suppliers and anesthetized by hypothermia until torpid and decapitated. All experiments involving the use of animals were performed in accordance with the guidelines of the National Institutes of Health and were approved by the USD Institutional Animal Care and Use Committee. Brainstems were transected at the levels of the trochlear and glossopharyngeal nerves and the cerebellum was removed as described previously leaving an isolated preparation of the pons (*Zheng et al., 2012*). Tissue was further transected down the midline for stimulation and recording of cranial nerves. The preparation was continuously bathed (2–4 ml/min) in physiological saline containing (in mM): 100 NaCl, 6 KCl, 40 NaHCO$_3$, 2.6 CaCl$_2$, 1.6 MgCl$_2$, and 20 glucose, which was oxygenated with 95% O$_2$/5% CO$_2$ and maintained at room temperature (22–24°C) at pH 7.6. Suction electrodes were used for stimulation and recording of cranial nerves. The unconditioned stimulus (US) was a twofold threshold single shock applied to the trigeminal nerve and the conditioned stimulus (CS) was a 100 Hz, 1 s train stimulus applied to the ipsilateral auditory nerve that was below threshold amplitude required to produce activity in the abducens nerve. Neural responses were recorded from the ipsilateral abducens nerve that innervates the extraocular muscles controlling movements of the eye, nictitating membrane and eyelid. The CS–US interval, defined as the time between the CS offset and the onset of the US, was 20 ms. The intertrial interval between the paired stimuli was 30 s. Preparations underwent the training procedure for a period of 15 min that consisted of 30 paired stimuli. In some cases, preparations were conditioned for one complete pairing session that consisted of 50 paired stimuli lasting 25 min in duration. Conditioned responses (CRs) were defined as abducens nerve activity that occurred during the CS and exceeded a twofold amplitude above the baseline recording level. Pseudoconditioned (Ps) control preparations received the same number of CS and US stimuli as conditioned preparations but these were explicitly unpaired using a CS–US interval randomly selected between 300 ms and 25 s. Naïve brainstems were presented with no stimuli and remained in the bath for the same time period as experimental preparations. Tissue samples for analysis were comprised of half the pons ipsilateral to the site of stimulation that contains the pontine portion of the eyeblink cranial nerve circuitry.

### Pharmacology

Preparations were incubated in zebularine (100 µM; Calbiochem), a cell-permeable cytidine analogue and DNA methylation inhibitor that acts by covalently bonding with DNMT, or RG108 (200 ng/µl; Axon Medchem), a non-nucleoside DNMT inhibitor that blocks the enzyme active site, applied to the bath for 1 hr prior to and throughout the training procedure.

## PCR and reverse transcriptase PCR (RT-PCR) for transcript expression

Total RNA was extracted from brainstem samples and an equal concentration was reverse transcribed using a 3' RACE adapter and M-MLV reverse transcriptase at 42°C for 1 hr as described previously (*Ambigapathy et al., 2013*). PCR was performed in two steps using primers specific for *tBDNF* (Table S2 in *Ambigapathy et al., 2013*). cDNA was amplified using the Accuprime *Pfx* polymerase system (Invitrogen). The primary PCR was carried out by using exon specific outer primers and 3' outer primers. Conditions for the PCR reaction were: initial denaturation at 94°C for 2 min, 25 cycles at 94°C for 30 s, 62°C for 30 s, 60°C for 30 s, 68°C for 2 min and final extension at 68°C for 10 min. The secondary PCR was carried out by using each primary PCR product as a template with exon specific inner primers and 3' inner primers. Conditions for the PCR reaction were: initial denaturation at 94°C for 2 min, 30 cycles at 94°C for 30 s, 60°C for 30 s, 68°C for 2 min and followed by final extension at 68°C for 10 min. The number of PCR cycles was optimized to maintain the amplification process within the linear range. Samples were confirmed to be free of DNA contamination by performing reactions without reverse transcriptase. PCR products were electrophoresed onto 2.0% agarose gels and stained with ethidium broide (0.5 µg/ml). Semi-quantitative analysis was performed as real-time PCR could not be used because transcript sequences were overlapping and the PCR product would be too long for optimal amplification. Images of the amplified products were acquired and the density of each band was background subtracted and measured with respect to $\beta$-actin using the InGenius Bio Imaging System (Syngene, Frederick, MD). The primers for $\beta$-actin were: Forward, 5'-AGGGAAATCGTGCGTGACAT-3'; Reverse, 5'-ATGCCACAGGATTCCATACC-3'. The resultant values were normalized as a percent of naïve values (%N).

To identify whether *tBDNF2a* is an intraexonic splice variant, genomic DNA was isolated from naïve brainstem samples using the Qiagen DNeasy mini kit (Qiagen). Total RNA was extracted from naive samples and reverse transcribed using oligo(dT)$_{20}$ and SuperScript III first strand (Invitrogen) following the manufacturer's protocol. These samples were confirmed to be free of DNA contamination by performing reactions without reverse transcriptase. Genomic DNA and cDNA were amplified using the Accuprime *Pfx* polymerase system and the PCR was carried out using primers that flanked the exon IV splice site (607–646 nt). The primer's were: forward primer, TACCCAAAGGCCAAC TGAAGCAATAC; reverse primer, TCTTCCCCTTTTAATGGTCAATGTACATAC. Conditions for the PCR reaction were: initial denaturation at 94°C for 2 min, 30 cycles at 94°C for 30 s, 58°C for 30 s, 68°C for 1 min and final extension at 68°C for 10 min. PCR products were electrophoresed onto 1.8% agarose gels and stained with ethidium broide (0.5 µg/ml). Amplified products were purified using the Purelink Quick gel extraction and PCR Purification Combo Kit (Invitrogen) and were sequenced by Functional Bioscience (Madison, WI).

## Bisulfite sequencing PCR (BSP)

Methylation status of the *tBDNF* protein coding sequence (exon IV) was analyzed using BSP. Genomic DNA was isolated from brainstem samples using the Qiagen DNeasy mini kit (Qiagen). DNA samples were treated with bisulfite reagent using the EZ DNA methylation-lightning kit (Zymo Research). Bisulfite-treated samples were amplified by PCR using primers that amplify the *tBDNF* coding sequence. Primers for BSP were designed using Methprimer software and are listed in *Table 1*. PCR reactions were carried out and amplified products were purified using the Purelink Quick gel extraction and PCR Purification Combo Kit (Invitrogen) and cloned into pGEMT-easy vectors (Promega). Selected colonies from each sample were sequenced by the Iowa State University DNA Facility or Functional Bioscience.

## Chromatin immunoprecipitation (ChIP) assays

ChIP assays were performed as previously described (*Ambigapathy et al., 2015*). Brainstem samples were minced on ice immediately after the conditioning procedures and incubated in 1% formaldehyde at 37°C for 10 min. The crosslinking reaction was stopped with 125 mM glycine for 5 min. The tissue was washed in ice-cold PBS containing protease inhibitors (1 mM phenylmethylsulfonyl fluoride, 1 µg/ml protease inhibitor) and homogenized with nuclei lysis buffer (50 mM Tris HCl, pH 8.0, 10 mM EDTA, 1% SDS). Chromatin was sheared by sonication into the range of 150–400 bp and then the lysates were centrifuged. The supernatant containing the sheared chromatin was diluted (1:10) with ChIP buffer (16.7 mM Tris, pH 8.0, 0.01% SDS, 1.1% Triton X-100, 167 mM NaCl, 1.2 mM

EDTA) and pre-cleared with a 50% suspension of salmon sperm DNA-saturated protein A/G agarose beads (Millipore) for 1 hr at 4℃ on a rotating platform. The agarose beads were pelleted by centrifugation and the supernatent immunoprecipitated using primary antibodies to: MeCP2 (Santa Cruz, 20700), Tet1 (ThermoFisher, GT1462), YB-1 (Cell Signaling, 9744), H3K4me3 and H3K27me3 (Cell Signaling, 9727 and 9733), CTCF (Cell Signaling, 3418), RNAPIISer2 (Millipore, 04–1571-I), Tet3 (Santa Cruz, 139186), or normal rabbit IgG (Santa Cruz, 2027) as a control, at 4℃ overnight with rotation. All of these antibodies were confirmed for their specificity by western blots of turtle and rat brain tissue as shown previously (*Ambigapathy et al., 2015*, their Figure 9) except for CTCF which is shown here in *Figure 6A*, and Tet3 (not shown). After overnight incubation, immune complexes were collected by incubating with 60 µl of 50% suspension of salmon sperm DNA-saturated protein A/G agarose beads for one hour at 4℃ with rotation. Agarose beads were pelleted by centrifugation and the immune complexes were washed once with low salt buffer (20 mM Tris, pH 8.0, 0.1% SDS, 1% Triton X-100, 2 mM EDTA, 150 mM NaCl), high-salt buffer (500 mM NaCl), LiCl wash buffer (0.25 LiCl, 10 mM Tris, pH 8.0, 1% deoxycholic acid, 1% IGEPAL CA-630, 1 mM EDTA), and twice with TE buffer (10 mM Tris, pH 8.0, 1 mM EDTA). Immunoprecipitated chromatin was eluted twice (1% SDS, 100 mM $NaHCO_3$) and 250 mM NaCl was added to reverse protein-DNA crosslinks by incubating the sample at 65℃ overnight. The samples were further incubated with proteinase K (100 µg/ml) for 1 hr at 50℃. DNA was isolated using phenol/chloroform/isoamyl alcohol followed by precipitation with ethanol, dried and dissolved in water. ChIP and input samples were amplified with *tBDNF* specific primers using quantitative real-time PCR (qPCR). *tBDNF* primers used for ChIP-qPCR are shown in *Table 2*. The qPCR was carried out by using SYBR green PCR master mix (Applied Biosystems). The fluorescence intensity of each amplicon was measured by the Step-One Plus real-time PCR system (Applied Biosystems). Conditions for qPCR were 95℃ for 10 min followed by 40 cycles at 95℃ for 15 s, and 60℃ for 1 min. Data from ChIP were normalized to input DNA for each sample ($\triangle Ct = Ct(sample) - Ct(input)$) and then further normalized relative to IgG control IP values ($\triangle\triangle Ct = \triangle Ct(experimental\ sample) - \triangle Ct(IgG\ control)$). The fold difference between the experimental sample and the negative IgG control was calculated by using $2^{(-\Delta\Delta Ct)}$ as described by *Mukhopadhyay et al. (2008)*. All reactions were performed in duplicates.

## Methylated DNA immunoprecipitation (MeDIP)

Genomic DNA was extracted from brainstem samples using the Qiagen DNeasy mini kit. DNA was sheared by sonication (150–400 bp) and 4 µg of fragmented DNA was used for the assays. DNA samples were denatured in a boiling water bath for 10 min and set in ice for 10 min. DNA was pre-cleared with protein A/G agarose beads for 1 hr at 4℃ then the supernatant was immunoprecipated with antibody against 5-hydroxymethylcytosine (Epigentek, A-1018) or normal IgG for 2 hr at 4℃ on a rotating platform. Immune complexes were collected by incubating with prewashed protein A/G agarose beads (60 µl) overnight at 4℃ with rotation. The immune complexes were washed once with low salt buffer, high-salt buffer, twice with lithium chloride wash buffer and TE buffer. The complexes along with input controls were then digested with proteinase K at 50℃ overnight. DNA was isolated using phenol/chloroform/isoamyl alcohol followed by precipitation with ethanol, dried and dissolved in water. MeDIP and input samples were amplified with *tBDNF* coding sequence primers (*Table 2*) using qPCR as described for the ChIP assays.

## Coimmunoprecipitation and western blot

Samples were homogenized in lysis buffer (20 mM Tris, pH 8.0, 1 mM EDTA, 1% Nonidet P-40, 150 mM NaCl, 10 mM $Na_4P_2O_7$, 5% glycine) with a protease (Roche) and phosphatase inhibitor (Sigma) cocktail. Protein samples were precleared with protein A/G agarose and supernatants were incubated in the primary antibodies or nonspecific rabbit or mouse IgG as a control at 4℃ for 2 hr. Protein A/G agarose was added to the protein samples and incubated at 4℃ overnight. Immunoprecipitated samples or IgG control samples were washed with ice-cold lysis buffer and dissociated by heating for 5 min in 2x loading buffer (4% SDS, 10% 2-mercaptoethanol, 20% glycerol, 0.004% bromophenol blue, 0.125 M Tris HCl, pH 6.8) and then subjected to SDS-PAGE. For all western blots and coimmunoprecipitation experiments both input protein and IgG controls were loaded at the same time. The same primary antibodies used for ChIP assays were also used for coimmunoprecipitation and/or western blotting. An antibody to *β*-actin (Millipore, 1501R) was used as a

loading control for western blotting. Proteins were detected by the ECL Plus chemiluminescence system (Amersham) or the Odyssey infrared imaging system (Li-Cor Biosciences) and quantified by computer-assisted densitometry.

### siRNA design and application

We designed two siRNAs targeting MeCP2 based on conserved regions of human MeCP2 mRNA (GQ896382.1) and the predicted amino acid sequence from the turtle *Chelonia mydas* (XM_007054052.1). The sense sequence for the siRNAs was 5′-UUGAUCAGGUAGACGUCGUAC UUUCCC-3′ corresponding to human MeCP2 619–645 nt and 5′-CGUCACCGUGAAGUCGAAAUCG UUGGG-3′ corresponding to 722–748 nt both present in human exon 4. The two duplex MeCP2 siRNAs were synthesized and purified (Integrated DNA Technologies, Coralville, IA). The siRNA to Tet1 was designed to target sequences in turtle corresponding to human exon 4 which is exclusive to the Tet1 isoform and is described elsewhere (*Ambigapathy et al., 2015*). The YB-1 siRNA was purchased from a commercial supplier (Cell Signaling, 6206). The siRNA against CTCF was designed using the predicted amino acid sequence for CTCF from the turtle *Chrysemys picta bellii* (XM_005312256.2) targeting 1187–1212 nt and synthesized (IDT). The sense sequence was 5′-CCAG-CAGAGAUACUUACAAACUGAA-3′. Finally, a commercially prepared scrambled negative control siRNA was also employed (Ambion, Silencer #1). The siRNAs were mixed with Lipofectamine RNAi-Max reagent (Invitrogen) in physiological saline and bath applied to preparations for 24 hr. The final concentrations used were: MeCP2, 200 nM; Tet1, 150 nM; YB-1, 200 nM; CTCF, 250 nM, and Silencer #1, 150 nM. After the elapsed time period, preparations underwent the conditioning procedure and were processed for further analysis.

### Statistical analysis

Data were analyzed with StatView software using a one-way ANOVA followed by a Fisher's LSD *post-hoc* test for paired comparisons or a Tukey's *post-hoc* test for multiple comparisons. Values are presented as means ± SEM. Sample size n represents the number of brainstem preparations. P values are determined relative to the naïve group except where noted. Significant differences were considered to be $p < 0.05$.

## Acknowledgements

We are indebted to Dr. Gail Mandel for valuable discussions during the course of this work.

## Additional information

### Funding

| Funder | Grant reference number | Author |
| --- | --- | --- |
| National Institutes of Health | NS051187 | Joyce Keifer |
| Internal departmental grant funds | | Joyce Keifer |

The funders had no role in study design, data collection and interpretation, or the decision to submit the work for publication.

### Author contributions

ZZ, GA, Data curation, Formal analysis, Validation, Investigation, Methodology, Writing—review and editing; JK, Conceptualization, Data curation, Formal analysis, Supervision, Funding acquisition, Validation, Investigation, Visualization, Methodology, Writing—original draft, Project administration, Writing—review and editing

### Author ORCIDs

Ganesh Ambigapathy, http://orcid.org/0000-0002-4491-8513
Joyce Keifer, http://orcid.org/0000-0002-5900-0414

## Ethics

Animal experimentation: All experiments involving the use of animals were performed in accordance with the guidelines of the National Institutes of Health and were approved by the USD Institutional Animal Care and Use Committee (protocol number, 08-06-14-17C).

## Additional files

### Supplementary files

• Supplementary file 1. Raw sequence data from splice site PCR and RT-PCR. Sequence data from PCR products generated from analysis of the splice site shown in *Figure 1C*. Sequence from the single band produced from genomic DNA is shown in sample 1. Data from cDNA produced two PCR bands whose sequences are shown for the upper band (sample 2) and the lower band (sample 3). 1indicates the splice site in sample 3.

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
