## [Decision Letter]

Thank you for submitting your article "MeCP2 regulates Tet1-catalyzed demethylation, CTCF binding, and learning-dependent alternative splicing of the *BDNF* gene" for consideration by *eLife*. Your article has been favorably evaluated by a Senior Editor and three reviewers, one of whom is a member of our Board of Reviewing Editors. The reviewers have opted to remain anonymous.

The reviewers have discussed the reviews with one another and the Reviewing Editor has drafted this decision to help you prepare a revised submission.

Summary:

In this manuscript the authors use an in vitro preparation from turtle to study the molecular mechanisms of BDNF regulation in a physiologically relevant conditioning paradigm. Previously the authors have reported the molecular characterization of BDNF transcription and splicing in this preparation, and here they explore how the methyl-DNA binding protein MeCP2 regulates an intraexonic BDNF splicing event both at rest and following conditioning. The data support a model in which MeCP2 recruits both the splicing factor YB1 and Tet1 to methylated regions of the BDNF coding regions under baseline conditions to promote expression of the truncated splice variant of BDNF. This complex is required for the conditioning-induced demethylation of BDNF, which is associated with induced binding of the insulator CTCF. Binding of CTCF is then subsequently required for loss of the truncated splicing event after conditioning.

The work is overall very well controlled and the reviewers are in agreement that the potential of the data to support a new model for how MeCP2 could control gene expression via alternative splicing is compelling. However, we had a vigorous debate over the significance of the particular splicing event under study, and some concerns were raised about the correlational nature of the serial knockdown studies. In the end, the reviewers agreed that the conceptual strengths of the manuscript outweigh the specific limitations assuming the following revisions can be made.

Essential revisions:

1) The manuscript must be rewritten to make the limited significance of the "splicing" event in BDNF more clear. To our knowledge, so far this kind of splicing event of BDNF gene has been described only in turtle and it has failed to be confirmed by RT-PCR in two major studies that examined BDNF splice isoforms in rodent and human. Furthermore, the biological meaning of this truncated protein isoform has not been studied. If the authors have data regarding the function of the truncated turtle BDNF protein it would substantially allay this concern if those data were added or referenced. However, if these experiments have not yet been done, we concur they are beyond the scope of the current study. In this case, substantial rewriting of the manuscript to make it clear that this particular splicing event serves to support the study of MeCP2 while backing off on the potential functional significance of the splicing, would be sufficient. The authors should also add a discussion of the implications that may arise from the species-specific nature of this splicing event. Specifically, what does this mean for the likelihood that the MeCP2 functions uncovered studying this regulatory process will be conserved in other species?

2) The authors need to clarify the model with respect to the direction of the splicing seen upon Tet siRNA. The authors show by siRNA that Tet1 recruitment is required for demethylation and demethylation presumably is required for CTCF binding. CTCF is required to inhibit the truncated splice variant with conditioning, however the authors also show that loss of Tet1 leads to loss of the truncated splice variant under baseline conditions. These findings do not fit neatly into a single model where methylation promotes the truncated variant by recruiting the MeCP2/YB1 complex. Does methylation of the BDNF gene change in the absence of Tet1? Does binding of YB1 to the BDNF gene change in the absence of Tet1?

3) Throughout the work, ANOVA with Fisher's (presumably Fisher's least significant difference) post hoc test was used for statistical analysis. The authors should justify the usage of Fisher's post hoc test, as it does not properly correct for multiple comparisons in case of more than 3 groups, and/or change the statistical analysis to correct for multiple comparisons where more than 3 groups are analyzed.

[Editors' note: further revisions were requested prior to acceptance, as described below.]

Thank you for resubmitting your article "MeCP2 regulates Tet1-catalyzed demethylation, CTCF binding, and learning-dependent alternative splicing of the *BDNF* gene in Turtle" for consideration by *eLife*. Your revised article has been reviewed by one peer reviewer, and the evaluation has been overseen by a Reviewing Editor and a Senior Editor.

We have only one remaining concern with the revised manuscript, and that regards the description of previous literature about BDNF splicing events in other species. Once this correction is made, the manuscript will be acceptable.

The major problem of the previous version of this manuscript was that the described splicing event of BDNF gene has been described only in turtle and the biological meaning of the putative truncated protein isoform has not been studied. Therefore, to increase the importance of the previous study the authors were asked to add data about evolutionary conservation of this intraexonic splicing of BDNF protein coding exon and to study/discuss also biological activity and meaning of this truncated BDNF protein.

The authors have added a paragraph in the Discussion about splicing of BDNF in different species, including intraexonic splicing of BDNF coding exon in human. Their interpretation of this finding as internal exonic splicing is correct. However, in human (but not in rodents) the intraexonic splicing is in the 5' untranslated region of the 5' extended region of the BDNF coding exon but not in the protein coding region of this exon. So according to current knowledge, BDNF transcripts generated by intraexonic splicing of protein coding region and encoding truncated BDNF protein are not present in rodents and human and there is no other species known other than turtle where such regulation takes place. This should be stated more clearly. In the present manuscript the reader can still get an impression that the splicing seen in turtle BDNF is conserved in other species.

---

## [Author Response]

*1) The manuscript must be rewritten to make the limited significance of the "splicing" event in BDNF more clear. To our knowledge, so far this kind of splicing event of BDNF gene has been described only in turtle and it has failed to be confirmed by RT-PCR in two major studies that examined BDNF splice isoforms in rodent and human. Furthermore, the biological meaning of this truncated protein isoform has not been studied. If the authors have data regarding the function of the truncated turtle BDNF protein it would substantially allay this concern if those data were added or referenced. However, if these experiments have not yet been done, we concur they are beyond the scope of the current study. In this case, substantial rewriting of the manuscript to make it clear that this particular splicing event serves to support the study of MeCP2 while backing off on the potential functional significance of the splicing, would be sufficient. The authors should also add a discussion of the implications that may arise from the species-specific nature of this splicing event. Specifically, what does this mean for the likelihood that the MeCP2 functions uncovered studying this regulatory process will be conserved in other species?*

After careful consideration of these comments, we have added in response a new paragraph to the beginning of the Discussion entitled, “*BDNF* gene structure and splicing across species” addressing these points. First, the reviewers/editors are correct that the biological significance of this splicing event and the resultant truncated BDNF isoform is currently unknown and this has been stated clearly in the paragraph and throughout the paper where referred to. The function of diverse isoforms generated by splicing has, generally, not been well studied and this is particularly true for *BDNF*.

There was also a concern about species-specificity of the splicing event and underlying mechanisms as they relate to MeCP2. The diversity of the *BDNF* gene among species (humans and rodents) in both its structure and splicing is summarized briefly in the new paragraph. Although intraexonic splicing events are unconventional and uncommon, they are documented to occur in other genes in humans and rodents as we briefly discuss. Regarding the reviewer’s specific comment about failure to confirm intraexonic splicing for the human *BDNF* gene, to my knowledge this was shown by Pruunsild et al. (2007). They showed that coding exon IX was comprised of subregions “a, b, c, and d” that generated transcript variants made up of different combinations of the regions (pg 398 and Figure 1 of their Results). We cited this information in our new text (subsection “*BDNF* gene structure and splicing across species”). However, if our interpretation of this finding as internal exonic splicing is incorrect, or if these data are unclear, the statement will be removed. The regulatory processes we have uncovered for the splicing event in turtle are, as documented in the paper, highly conserved compared to mammals. That is, Tet-mediated DNA methylation controls binding of MeCP2-YB-1 and CTCF in a highly coordinated manner related to conditioning. We now emphasize the point at the beginning of the Discussion that we are using this *BDNF* splicing event as a model to study mechanisms of activity-dependent splicing related to MeCP2 function.

*2) The authors need to clarify the model with respect to the direction of the splicing seen upon Tet siRNA. The authors show by siRNA that Tet1 recruitment is required for demethylation and demethylation presumably is required for CTCF binding. CTCF is required to inhibit the truncated splice variant with conditioning, however the authors also show that loss of Tet1 leads to loss of the truncated splice variant under baseline conditions. These findings do not fit neatly into a single model where methylation promotes the truncated variant by recruiting the MeCP2/YB1 complex. Does methylation of the BDNF gene change in the absence of Tet1? Does binding of YB1 to the BDNF gene change in the absence of Tet1?*

To more fully clarify our model of splicing with regard to the Tet1 siRNA data, we performed additional experiments and have included them in a new Figure 5. Additionally, in light of these data, a revised discussion of the MeCP2-Tet1 interaction is included.

New experiments on the mechanisms of splicing inhibition by the Tet1 siRNA are reported in the Results (subsection “Inhibition of Tet1 blocks MeCP2-YB-1 *tBDN*F binding and splicing”) and in a new Figure 5. In the absence of Tet1, methylation as measured by BSP does not change compared to normal naïve preparations (Figure 5), but levels of 5hmC examined by MeDIP greatly increase (Figure 5). This has been observed in studies of mammals using Tet1 knockdown (e.g., Feng et al. 2015, PMID 25774451). Importantly, we discovered that in conditions of Tet1 siRNA, there is substantial inhibition of MeCP2 and YB-1 *tBDNF* binding (Figure 5), which explains why splicing is blocked (Figure 3). Correspondingly, we also observed that when Tet1 is absent, Tet3 binds in its place. Therefore, the data indicate that MeCP2-YB-1 is inhibited from binding in the absence of Tet1, even though the DNA is methylated.

These data further illustrate the highly complex nature of the MeCP2-Tet1 interaction. To discuss this more fully, a new paragraph has been added to the Discussion (subsection “Impact of MeCP2 loss of function on DNA methylation and chromatin”). Here, we note that knockdown of either MeCP2 or Tet1 negatively impacts the function of the other suggesting they are critical functional partners. We conclude that together they may regulate the overall content of 5mC and 5hmC in dynamically modulated genes (subsection “Impact of MeCP2 loss of function on DNA methylation and chromatin”, last paragraph) and, in the case of *tBDNF*, regulate splicing.

*3) Throughout the work, ANOVA with Fisher's (presumably Fisher's least significant difference) post hoc test was used for statistical analysis. The authors should justify the usage of Fisher's post hoc test, as it does not properly correct for multiple comparisons in case of more than 3 groups, and/or change the statistical analysis to correct for multiple comparisons where more than 3 groups are analyzed.*

All of the multiple comparisons were now corrected using a Tukey’s post-hoc test. Some of the *p* values were changed, but there were no changes in what was previously significant in the original manuscript. Paired comparisons (2 groups) were still made with the Fisher’s test. These changes are stated in the Methods for Statistical Analysis.

[Editors' note: further revisions were requested prior to acceptance, as described below.]

*[…] The authors have added a paragraph in the Discussion about splicing of BDNF in different species, including intraexonic splicing of BDNF coding exon in human. Their interpretation of this finding as internal exonic splicing is correct. However, in human (but not in rodents) the intraexonic splicing is in the 5' untranslated region of the 5' extended region of the BDNF coding exon but not in the protein coding region of this exon. So according to current knowledge, BDNF transcripts generated by intraexonic splicing of protein coding region and encoding truncated BDNF protein are not present in rodents and human and there is no other species known other than turtle where such regulation takes place. This should be stated more clearly. In the present manuscript the reader can still get an impression that the splicing seen in turtle BDNF is conserved in other species.*

There was one remaining concern about the paragraph that was added to the Discussion about *BDNF* splicing across species. It was pointed out by the reviewers that, while the event for human *BDNF* described by Pruunsild et al. (2007) is internal exonic splicing, it does not occur within the protein coding region. Therefore, in response, we have removed this statement entirely. We also removed the statement regarding the Aid et al. (2007) study since the extended transcription initiation site does not appear to be translated to protein. Additionally, some of the text in that same paragraph was edited for clarity (subsection “*BDN*F gene structure and splicing across species”).